# RDAR: REWARD-DRIVEN AGENT RELEVANCE ESTIMATION FOR AUTONOMOUS DRIVING

**Carlo Bosio**[*]
UC Berkeley
c.bosio@berkeley.edu

**Greg Woelki**
Zoox Inc.
gwoelki@zoox.com

**Noureldin Hendy**
Zoox Inc.
nhendy@zoox.com

**Nicholas Roy**
Zoox Inc.
nroy@zoox.com

**Byungsoo Kim**
Zoox Inc.
bykim@zoox.com

## ABSTRACT

Human drivers focus only on a handful of agents at any one time. On the other hand, autonomous driving systems process complex scenes with numerous agents, regardless of whether they are pedestrians on a crosswalk or vehicles parked on the side of the road. While attention mechanisms offer an implicit way to reduce the input to the elements that affect decisions, existing attention mechanisms for capturing agent interactions are quadratic, and generally computationally expensive. We propose RDAR, a strategy to learn per-agent relevance — how much each agent influences the behavior of the controlled vehicle — by identifying which agents can be excluded from the input to a pre-trained behavior model. We formulate the masking procedure as a Markov Decision Process where the action consists of a binary mask indicating agent selection. We evaluate RDAR on a large-scale driving dataset, and demonstrate its ability to learn an accurate numerical measure of relevance by achieving comparable driving performance, in terms of overall progress, safety and performance, while processing significantly fewer agents compared to a state of the art behavior model.

## 1 INTRODUCTION

Humans, when driving, do not pay equal attention to all agents around them (e.g., other vehicles, pedestrians). Transformer-based attention models offer the promise of attending only to relevant components of the input, but existing attention models are typically quadratic in the size of the input space. Driving models encounter hundreds of input tokens, leading to substantial computational complexity and latency Harmel et al. (2023); Huang et al. (2024); Baniodeh et al. (2025).

In autonomous driving, there is a tension between the limited available compute resources and the desire to take advantage of scaling laws, large models, and test-time compute. Having access to numerical per-agent relevance scores would not only improve the interpretability of large driving models, but also allow compute resources to be prioritized for the features that are most important. In fact, when agents and other scene elements are represented explicitly as tokens, reasoning about interactions between these tokens (typically through self-attention or graph neural network operations) is quadratic and difficult to reduce using low-rank or other approximations that work well for long-sequence data. Reducing the number of tokens under consideration provides quadratic improvements in FLOPs used.

In this work, we introduce RDAR (Reward-Driven Agent Relevance), through which we quantify agent relevance through a learned approach. The basic intuition is that if an agent is not relevant towards the driving decisions of the controlled vehicle, then its absence would not change the controlled vehicle's driving behavior significantly. Thus, we quantify per-agent relevance by learning which agents can be masked out from the controlled vehicle's planner input while maintaining a good driving behavior. We formulate agent selection as a reinforcement learning (RL) problem where an

---

[*]Work completed while the author was an intern at Zoox, Inc.

Figure 1: Example visualizations showing agent relevance assigned by our method. The top-$k$ relevant agents are labeled with a colored dot hovering over them. As shown by the scale in the bottom left of each image, red corresponds to higher relevance and light blue corresponds to lower relevance. a) attending to cyclist while turning left, b) attending to pedestrian during stop. The controlled vehicle is in black.

action is a binary mask indicating which agents to include in the driving policy input, and which not to. The RDAR scoring policy is trained in the loop with a frozen, pre-trained driving policy and a simulator. At each time step, based on the relevance scores, an agent mask is fed to the driving policy, making the controlled vehicle blind to the lower score agents. As it will be clear from the following sections, this is not a binary classification problem over agents due to the underlying system dynamics (e.g., not observing an agent now could lead to a collision later) and to the unavailability of ground truth labels. Some examples of relevance (color-coded) computed by our method are shown in Fig. 1. Our main contributions are:

- A novel reinforcement learning formulation for agent relevance estimation;
- A sampling-based mechanism for agent selection that enables efficient training and inference;
- A comprehensive evaluation showing that we can maintain driving performance while processing only a handful of surrounding agents.

## 2 RELATED WORK

Learning object ranking is a long standing problem in deep learning and typically requires human-labeled data Cohen et al. (1997); Burges et al. (2005); Jamieson & Nowak (2011). In fields such as autonomous driving, a manual ranking process can be not straightforward, and require large amounts of labeled data. Input attribution Sundararajan et al. (2017) is a family of post-hoc analysis methods attempting to pinpoint which parts of the input are most responsible for a prediction. Attribution methods mostly focus on leave-one-out (LOO) approaches Liu et al. (2024), where chunks of the input are individually removed, or masked, and are correlated with changes in model outputs. Attention weights are sometimes believed to correlate with feature importance, but this correlation has been shown to be weak Jain & Wallace (2019).

Another relevant field in Machine Learning research is feature selection, i.e. methods that aim to identify a subset of the most informative input variables for a model Li et al. (2017). Recently, Reinforcement Learning–based feature selection approaches have been proposed Liu et al. (2021); Rasoul et al. (2021). However, despite leveraging RL machinery, these methods generally do not address sequential decision-making problems: the objective is typically minimizing the prediction error in a supervised learning setting. Moreover, the action space is typically the power set of the feature set, making it combinatorial and rapidly intractable as the number of features grows. Our method allows to quantify agents' importance *across time* through a formulation that scales linearly with the number of agents in a driving scene, instead of combinatorially.

Ranking agents in a driving scene based on their relevance is useful for both offline and online applications. Current autonomous driving systems quantify the relevance of surrounding agents either through fast, heuristic-based modules (e.g. based on euclidean distance), or learned models trained through supervised learning. Some approaches have been proposed for the supervision of these models, and they predominantly focus on LOO strategies coming from the attribution literature for agent prioritization Refaat et al. (2019), selective prediction Tolstaya et al. (2021), or offline introspection Cusumano-Towner et al. (2025). While these LOO approaches provide insights into individual agent contributions, they have some limitations. First of all, a change in driving behavior,

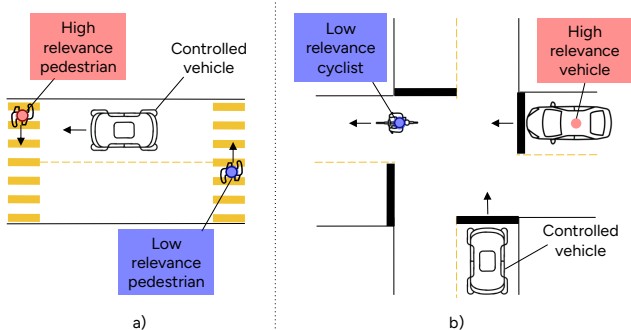

Figure 2: Example driving scenarios highlighting agent relevance. In a), the pedestrian crossing in front of the controlled vehicle is highly relevant, while the one behind is not. In b), the bike just went through the intersection and is not relevant anymore, while the car inching into the intersection is highly relevant.

captured by a shift in predicted action, represents a *different* driving behavior, but not necessarily a *worse* one. Second, these methods require multiple forward passes through a model (a planner in this case). Third, they do not capture the temporal dependencies caused by system dynamics. In this work, we propose a reward-driven method trained through reinforcement learning to estimate agent relevance.

RDAR computes per-agent relevance through just one forward pass, and is reward-driven instead of supervised through ground truth labels. Reinforcement learning (RL) approaches have shown promising results in autonomous driving applications Kiran et al. (2021), and are being integrated in production and deployed in real-world systems. Several works have shown large scale urban driving through the use of both RL and human-collected real world driving data Harmel et al. (2023); Gulino et al. (2023); Lu et al. (2023). In this work we build upon an existing learned behavior model, and train a scoring policy with closed-loop RL through a novel formulation for agent selection.

## 3 PROBLEM SETUP

We wish to learn a policy $\pi_\theta^R$ assigning a *relevance score* to each agent in the driving scene based on its influence on the driving behavior of the controlled vehicle ($\theta$ denotes learnable parameters). We assume a pre-trained driving policy $\pi^D$ mapping scene information to driving actions is available. We also assume that, associated with the policy $\pi^D$, there is a reward function $r$ encoding some notion of good driving behavior. The RDAR scoring policy $\pi_\theta^R$ is trained in closed loop with the (frozen) driving policy $\pi^D$ and a driving simulator.

Formally defining a notion of agent relevance is not straightforward. However, human drivers have a good intuitive concept of such notion, which allows them to pay selective attention to surrounding agents. With reference to Fig. 2a, a pedestrian crossing in front of the controlled vehicle is a highly relevant agent, because its presence its presence means that the controlled vehicle must come to a stop and yield instead of driving through a crosswalk. At the same time, the pedestrian crossing behind the driver has low relevance. Similar considerations are true for the intersection scenario of Fig. 2b. The vehicle inching into the intersection has high relevance, because its presence means that the controlled vehicle must stop and yield. Instead, the cyclist who just passed through the intersection should not affect the controlled vehicle behavior. Therefore, we can say that an agent is relevant if hypothetically removing it from a driving scenario would cause the controlled vehicle to have a different behavior.

**Markov Decision Process formulation**   Following the above intuition, we formulate the agent relevance estimation problem as a Markov Decision Process (MDP). The policy $\pi_\theta^R$ outputs per-agent relevance scores, which can be interpreted as logits of a categorical distribution. If agent $i$ is sampled from this *relevance distribution*, it gets processed by the driving policy $\pi^D$, otherwise it is masked out and ignored by $\pi^D$. Given a hyperparameter $k \in \mathbb{N}$, an action is then a subset of $k$ surrounding

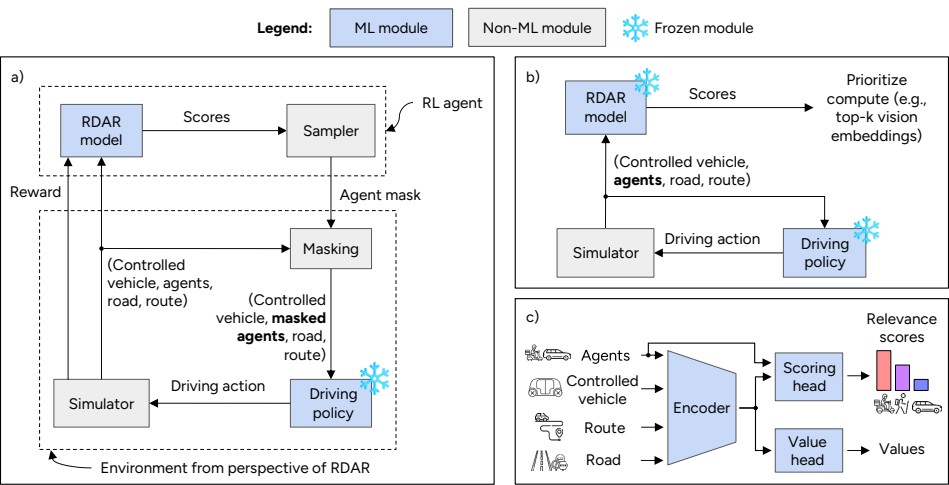

Figure 3: a) Block diagram of the system at training time. b) Block diagram of the system at deployment time. c) RDAR model structure.

agents, or a $k$-sample, to be processed by $\pi^D$. Our goal is thus to learn $\pi_\theta^R$ such that the return is maximized in expectation. Inaccurate relevance scores would make the driving policy blind to important agents in the scene, leading to low reward behaviors (e.g., collisions). The MDP setup for this process is defined by a standard tuple $(S, A, r, P, \mu_0)$, where:

- $S$ is the state space, including the controlled vehicle state, surrounding agent states (expressed in the controlled vehicle reference frame), road network and route information (see Fig.3c);
- $A = \{0, 1\}^N$ is the action space, consisting of binary masks of size $N$ (number of agents) indicating which agents to include in the planning input. The logits of the action distribution are the relevance scores (details in the following sections);
- $r$ is the reward function encoding good driving behavior. This is ideally the same reward or scoring function accompanying $\pi^D$;
- $P$ is the transition probability function associated to the environment. Note that the environment, from the perspective of the reinforcement learning agent $\pi_\theta^R$, consists of $\pi^D$ and the actual driving environment altogether (see Fig.3a);
- $\mu_0$ is the initial state distribution.

The nature of the actions space makes this problem similar to a contextual multi-armed bandit (CMAB) Lu et al. (2010), with the subtle difference that in this case the action changes the context, which in CMABs is assumed to be independent of the action.

**Agent Selection Mechanism**  At each timestep $t$, the RDAR policy $\pi_\theta^R$ outputs a vector of per-agent relevance scores $\phi_t = [\phi_1, \phi_2, \ldots, \phi_N]$. At deployment time, the top-$k$ scoring agents are selected, while at training time, agents are randomly sampled to encourage exploration. For sampling, the scores are converted to a categorical distribution $p_t = [p_1, p_2, ..., p_N]$ over the binary agent selection action space through a softmax, where

$$p_i = \frac{\exp(\phi_i)}{\sum_{j=1}^{N} \exp(\phi_j)}. \tag{1}$$

Drawing one sample from this distribution corresponds to selecting one agent. If we draw exactly one sample, the probability of agent $i$ being selected is $p_i$. We can thus get a $k$-sample by drawing $k$ samples sequentially, without replacement, by renormalizing the probabilities at each step. We denote an agent $k$-sample as

$$a = (a_1, a_2, \ldots, a_k), \ \ a_i \in \{1, ..., N\}, \tag{2}$$

where each component $a_i$ is the integer index corresponding to the selected agent. Note that this notation and an $N$-dimensional binary vector are equivalent. Then, the probability of selecting an

ordered sample of agents without replacement is

$$P(a_1, \ldots, a_k) = \prod_{i=1}^{k} \frac{p_{a_i}}{1 - \sum_{j=1}^{i-1} p_{a_j}}, \tag{3}$$

where the denominators are the renormalization terms. Note that although we describe the sampling process as sequential, the Gumbel top-$k$ trick enables efficient, single-step sampling without replacement Jang et al. (2016); Kool et al. (2019). The trick consists of perturbing the distribution $p_t$ with a Gumbel distribution, and greadily selecting the top-$k$ elements:

$$(a_1, ..., a_k) = \arg \text{ top-}k \ \{ p_t - \log(-\log U) \}, \quad U \sim \text{Uniform}(0, 1) \tag{4}$$

By applying the logarithm to equation 3, we can compute the log-likelihood of the $k$-sample:

$$\log P(a_1, \ldots, a_k) = \sum_{i=1}^{k} \log p_{a_i} - \sum_{i=1}^{k} \log \Big( 1 - \sum_{j=1}^{i-1} p_{a_j} \Big). \tag{5}$$

Since the scores are the output of our model $\pi_\theta^R$, equation 5 is exactly what we need for policy gradient updates in an RL framework. Once a $k$-sample is picked, only those $k$ agents are processed by the driving policy $\pi^D$. The action output by the driving policy is then applied to the simulator, and the overall state is updated. The simulator also produces the reward signal $r_t$ for RDAR. The process is then repeated.

**Reinforcement learning framework** We train the policy using an off-policy actor–critic framework with V-trace corrections Espeholt et al. (2018). The loss function combines four components:

$$\mathcal{L} = \mathcal{L}_{\text{policy}} + \lambda_c \mathcal{L}_{\text{critic}} + \lambda_e \mathcal{L}_{\text{entropy}} + \lambda_s \mathcal{L}_{\text{smoothing}}. \tag{6}$$

The four components are respectively policy gradient loss, critic loss, entropy regularization loss, and action smoothing loss. Their exact expressions are:

$$\mathcal{L}_{\text{policy}} = -\mathbb{E}_t \left[ \rho_t \, \log \pi_\theta^R(a_t \mid s_t) \, \hat{A}_t^{\text{v-trace}} \right], \tag{7}$$

$$\mathcal{L}_{\text{critic}} = \mathbb{E}_t \left[ \big( V_\theta(s_t) - V_t^{\text{target}} \big)^2 \right], \tag{8}$$

$$\mathcal{L}_{\text{entropy}} = \mathbb{E}_t \left[ -\sum_{i=1}^{N} \pi_\theta^R(i \mid s_t) \log \pi_\theta^R(i \mid s_t) \right], \tag{9}$$

$$\mathcal{L}_{\text{smooth}} = \mathbb{E}_t \left[ \sum_{i=1}^{N} \| \pi_\theta^R(i \mid s_t) - \pi_\theta^R(i \mid s_{t-1}) \|^2 \right], \tag{10}$$

where $\rho_t = \frac{\pi_\theta^R(a_t | s_t)}{\mu(a_t | s_t)}$ are clipped importance weights, $a_t$ is the agent $k$-sample at time step $t$ as in equation 2. The $\hat{A}_t^{\text{v-trace}}$ and $V_t^{\text{target}}$ terms are computed from the driving reward following Espeholt et al. (2018). The log-likelihood term in equation 7 is computed as in equation 5. The entropy and smoothing loss components, are calculated on the logits directly and do not depend on the $k$-sample $a_t$. The entropy component favors uniformity in the relevance scores, and therefore encourages exploration. The action smoothing component encourages the scores corresponding to the same agent to be consistent across time. Note that, even if the reward function used to train $\pi_\theta^R$ is the same as the one used for the underlying driving policy $\pi^D$, the corresponding value functions have different meanings. In fact, the expectation operators in the two cases differ due to the different action spaces.

Some example scenarios showing the time consistency of the RDAR relevance scores can be seen in Appendix B. It is possible that fewer than $N$ agents are physically present in a scene at a given time, in which case the loss terms corresponding to non-existing agents are masked. Finally, $\lambda_c, \lambda_e, \lambda_s$ are hyperparameters weighing the loss terms, selected empirically. The driving reward used in our MDP setup, and in equation 7 and equation 8, is a combination of dense rewards as well as done signals that are associated with sparse rewards. The overall objectives are progress along the desired route while not colliding with other agents and adhering to basic traffic rules.

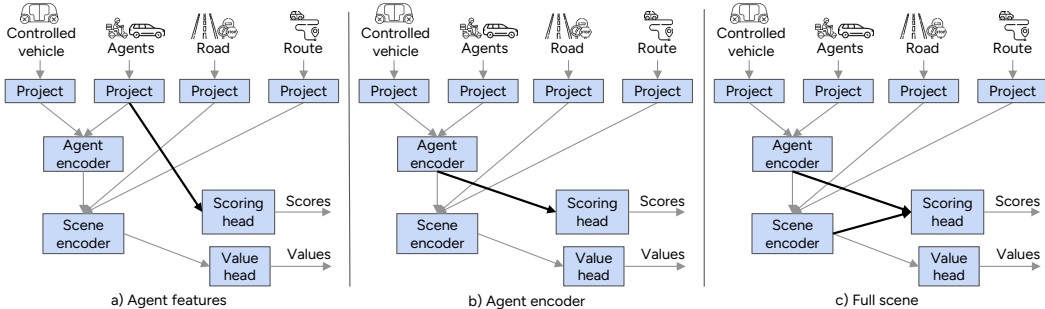

Figure 4: Different model architectures tested. Left: the scoring head consumes directly agent features projected through a linear layer. Middle: the scoring head consumes the output of the agent encoder, a transformer block cross attending among agents. Right: the scoring head consumes full scene information encoded into a latent embedding from the scene encoder, and attending back to agent tokens.

## 4 IMPLEMENTATION

**Architecture** The RDAR model architecture consists of three main components. An encoder module processes the full scene context (controlled vehicle state, surrounding agent states, road and route information) and computes embeddings. There are two heads: a scoring head, mapping embeddings to agent relevance scores, and a value head, approximating the value function. Note that the value function for the relevance scoring policy has a different meaning than the value function for the driving policy, since the expectation is over all possible $k$-samples rather than all possible driving actions. The three options for the scoring head make use of embedding of varying depth from the encoder. The first option (Fig. 4a) just uses the features from the agent projection layer. In this case, only agent state information is fed to the scoring head. The second option (Fig. 4b) uses the embeddings output by the agent encoder module. In this case, the embeddings also encode agent interactions. The third option (Fig. 4c) uses the output from the scene encoder, and reprojects it back to the agent level through a transformer block. In this case, all information from the driving scene is used to compute agent relevance. The value head is kept the same for the three architectures, and uses all available information.

**Training details** The training-time agent sampling procedure described in Section 3 is done through the Gumbel top-$k$ trick (Kool et al. (2019)), natively used, for instance, in the JAX's implementation of the categorical distribution Bradbury et al. (2018). Also at training time, the number $k$ of agents sampled is randomized to make sure the model learns actual relevance scores and not only to differentiate between top-$k$ and non top-$k$ agents. We uniformly sample a different value $k$ per driving scenario. Having truthful relevance scores also allows, at inference time, to vary the number of $k$ agents to process downstream, based for example on the scenario complexity. A simple variable $k$ sampling strategy can be a top-$p$ sampling based on the normalized scores, as in Holtzman et al. (2019). More complicated sampling strategies based on scene complexity can be considered and are left for future work. To achieve scale, we use a distributed, asynchronous reinforcement learning infrastructure similar to IMPALA Espeholt et al. (2018). We found these hyperparameter values to give the best performances: learning rate $2 \cdot 10^{-5}$, $\lambda_c = 0.1$, $\lambda_e = 0.2$, $\lambda_s = 0.05$. Additional details about the sensitivity to the loss hyperparameters can be found in Appendix A. No sampling happens at deployment time, and and the top-$k$ agents are selected greedily (analogous to selecting the argmax action in standard RL).

## 5 EXPERIMENTAL SETUP

**Datasets** We train and evaluate our approach on large-scale proprietary datasets consisting of ten-second long scenarios of real world, diverse urban driving. The training dataset contains around two million scenarios, while the evaluation dataset contains twelve thousand. The data were collected across Las Vegas (LV), Seattle (SEA), San Francisco (SF), and the campus of the Stanford Linear

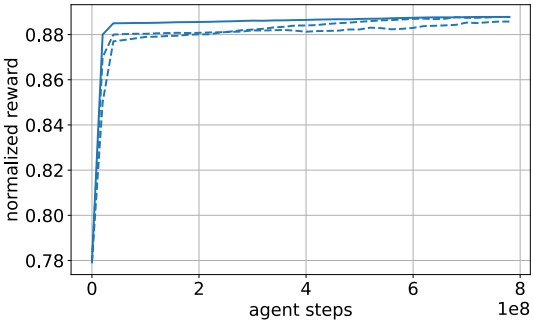

Figure 5: Example learning curves obtained with three different random seeds. The solid line represents the highest performing experiment. The evaluation is done every 20M agent steps, and during the evaluation $k$ is fixed to 10.

Accelerator Center (SLAC). Note that Las Vegas, Seattle, and San Francisco represent dense urban settings, while SLAC is a suburban academic campus. Training and validation data are not separated by location, so a given validation example may have training examples from the same location at other times. Scenarios are sampled at $5\,\mathrm{Hz}$ from real-world driving logs.

**Metrics**   To quantitatively evaluate our method, we use standard driving metrics which we compute on the 12k scenario evaluation set. In these evaluations, the candidate relevance scoring policies and baselines are used in closed-loop as filters, with the driving policy $\pi^D$ processing only a subset $k$ of all the agents present at any one time. At evaluation, we select the top-$k$ scoring agents greedily (analogously to selecting an action through argmax in standard RL). We use the following metrics:

1. *Collisions [%]*: percentage of scenarios in which a collision occurs (lower is better);
2. *Traffic light [%]*: percentage of scenarios in which a traffic light is violated (lower is better);
3. *Stop line [%]*: percentage of scenarios in which a stop line is skipped (lower is better);
4. *Off-road [%]*: percentage of scenarios in which the vehicle drives off-road (lower is better);
5. *Comfort*: metric combining four motion aspects – forward/backward acceleration, turning acceleration, and how suddenly or abruptly these accelerations change. These are weighted, averaged, then converted to a 0-1 score where 1 means smooth driving and 0 means jerky, uncomfortable motion (higher is better).
6. *Progress ratio*: relative progress along the route with respect to the ground truth human log;
7. *Complexity*: computation required by the scoring method as a function of the number $N$ of agents in a scene.

**Baselines**   We compare RDAR to other scoring strategies. The evaluation is done in closed-loop, using these strategies to pick the top-$k$ agents to be processed by $\pi^D$:

- Closest-$k$ selection: Select the $k$ closest agents to the controlled vehicle – equivalent to agents' relevance scores being inversely proportional to their distance to the controlled vehicle;
- Random-$k$ selection: Randomly select $k$ agents from the scene – equivalent to agents' relevance scores being drawn uniformly at random;
- Attribution-based scoring: scores obtained via input attribution Cusumano-Towner et al. (2025). At each timestep, the procedure is as follows: the pre-trained driving policy $\pi^D$ is evaluated $N + 1$ times (one with full scene information and one with each individual agent omitted in turn). For each agent, the Jensen–Shannon divergence between the action from its masked-out pass and the nominal full-scene pass is computed.

The overall performance of these baselines when varying $k$ is shown in Fig. 6. When $k = N$ no agents are masked out from the driving policy.

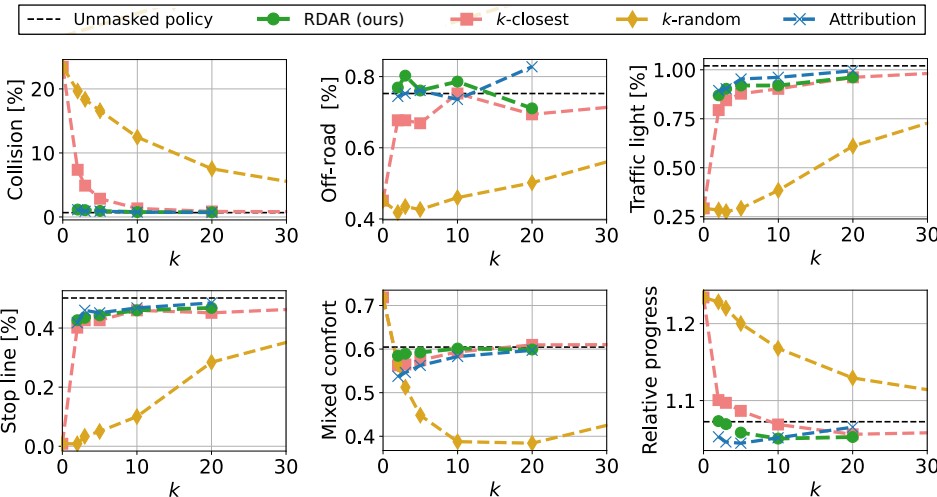

Figure 6: Metrics of best RDAR model and baselines against number $k$ of selected agents in closed-loop. All the closed-loop metrics for RDAR are close to the full policy baseline, supporting the claim that we are able to learn true agent relevance. In this case, the RDAR model is the version processing full scene information (see Fig. 4c).

## 6 RESULTS

We trained our RDAR models on a compute node with 32 A100 GPUs, for a total of roughly 200 GPU hours. This corresponds to 1 million agent steps. As shown by the training curves (ADD), the evaluation performance reaches 90% within the first 10% of the training. Some example training curves from RDAR are shown in Fig. 5.

**Quantitative results** We report plots showing the trends of the reference metrics when varying the value $k$ in Fig. 6. As expected, all the metrics seem to converge to the baseline's when increasing the number of agents processed by the driving policy, with diminishing returns. We can see that RDAR has comparable performances to the attribution method, and requires only a fraction of the computation time ($O(1)$ instead of $O(N)$). While we have not implemented our RDAR model on our full-scale autonomous vehicle, we found that a vectorized implementation of the attribution method (i.e. $N + 1$ forward passes of the driving policy) is on average 10x slower than RDAR on the evaluation set. RDAR is also able to drive with a fraction of percent performance regressions compared to the nominal, full policy, while processing significantly fewer agents. In fact, an RDAR model used in closed-loop with $k \geq 10$ already enables comparable driving performances to the nominal unmasked policy while consuming an order of magnitude fewer agents. With increasing $N$, as expected the collision rates decrease, the comfort scores increase (due to reduced flickering of agent selection). However, traffic infraction rates seem to have an upward trend. This makes sense, as for instance driving on empty roads would make it very easy to avoid infractions, and driving in heavy traffic makes it harder due to the higher risk of collisions.

Tab. 1 shows a comparison relative to the $k = 10$ case. The three different architectures proposed in Fig. 4 show comparable performances when used in closed-loop with the driving policy. The model using full scene information (Fig. 4c) leads to the least collisions and higher comfort score. There seems to be a trade-off between collision avoidance and the metrics related to traffic rules, which can be explained by the fact that avoiding collisions could require the controlled vehicle to do an infraction (e.g. driving off-road). In fact, while full scene information allows to minimize collisions, it does that at the cost of slightly higher infraction rates. Similarly, the baseline, unmasked driving policy has higher infraction rates than most masked alternatives. It is also interesting to see that the random scoring policy outperforms all the others when it comes to traffic rules (off-road, traffic lights, stop lines), which comes, as expected, at the cost of much higher collision rates. Scores computed using only agent features (Fig. 4a) or attending to the controlled vehicle state (Fig. 4b) achieves good closed-loop driving and requires fewer FLOPs compared to using full scene information (Fig.

|  | Collisions [%] | Off-road [%] | Traffic light [%] | Stop line [%] | Comfort | Rel. progress | Complexity |
|---|---|---|---|---|---|---|---|
| RDAR, Agent features (Fig. 4a) | 0.94 | 0.71 | 0.97 | 0.44 | 0.57 | 1.06 | $O(1)$ |
| RDAR, Agent encoder (Fig. 4b) | 0.89 | 0.70 | 0.89 | 0.43 | 0.57 | 1.06 | $O(1)$ |
| RDAR, Full scene (Fig. 4c) | 0.77 | 0.79 | 0.92 | 0.46 | 0.60 | 1.05 | $O(1)$ |
| Closest-$k$ | 1.34 | 0.75 | 0.90 | 0.46 | 0.59 | 1.07 | $O(1)$ |
| Random-$k$ | 12.5 | 0.46 | 0.38 | 0.10 | 0.39 | 1.17 | $O(1)$ |
| Attribution | 0.75 | 0.74 | 0.96 | 0.47 | 0.58 | 1.05 | $O(N)$ |
| Baseline (no filter) | 0.68 | 0.75 | 1.02 | 0.50 | 0.61 | 1.07 | — |

Table 1: Closed-loop metrics of different methods with $k = 10$ agents. Using only agent features or reasoning about agent interactions leads to good closed-loop driving performance, and requires fewer FLOPs compared to full scene, which on the other hand achieves lower collision rate.

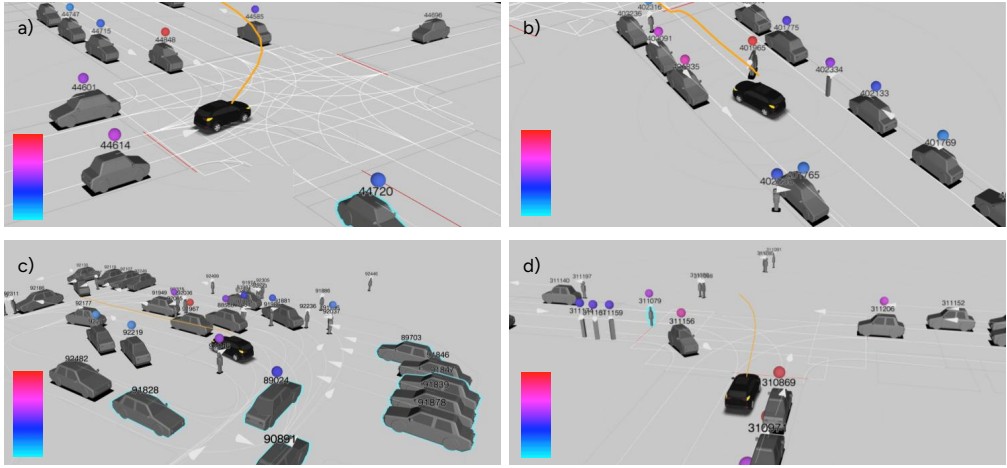

Figure 7: Example visualizations showing agent relevance assigned by our method. The top-$k$ relevant agents are labeled with a colored dot hovering over them. As shown by the scale in the bottom left of each image, red corresponds to higher relevance and light blue corresponds to lower relevance. The controlled vehicle is in black. Agent importance seems to be aligned with human intuition, highlighting high-risk pedestrians, other cars the controlled vehicle must yield to, agents in close proximity.

4c). On the other hand, full scene information enables enhanced situational awareness and lower collisions. Therefore, the specific RDAR architecture can be chosen based on application needs in terms of memory, real-time compute requirements, or score accuracy.

**Qualitative results**  We also report some visualizations from some closed-loop evaluation rollouts (the same used to compute the aggregate metrics) in Fig. 7. The scenes represented are challenging, cluttered driving scenes in which incorrect relevance quantification would lead to wrong masking and bad behaviors. In the figures, a colored dot hovers over the top-10 agents. These are the agents that are being processed by the driving policy, while the other are ignored. As shown in the color scale in the bottom left of each figure, the red color corresponds to high relevance score, the light blue color corresponds to lower relevance (still, within the top-10). These examples are from our best full-scene scoring policy (Fig. 4c). We can see that the agent importance assigned by our model is aligned with human intuition, and highlights high-risk pedestrians, other cars the controlled vehicle must yield to, agents in close proximity.

# 7 DISCUSSION AND CONCLUSIONS

Our model provides insights into which agents influence driving decisions, enabling better understanding of planner behaviors. It can also inform how to allocate costly computation in a principled way–for example, running joint trajectory prediction or computing vision embeddings only for the most relevant agents.

This work opens several interesting directions. First, similar relevance-scoring methods could be applied to other components of the driving scene, such as road information. A challenge here is the potential for distribution shifts when masking inputs; in our case such effects are mild, since driving scenarios remain in-distribution regardless of the number or position of agents, but investigating mitigation strategies is important. Second, the scoring policy's action space could be expanded beyond masking. Instead of excluding agents, the prioritization module could be trained to trigger targeted computation on selected agents, such as expensive vision embeddings, so that enhanced representations directly translate into downstream gains like improved driving rewards. Most related to this vision is conditional computation applied to LLMs, such as what proposed by Raposo et al. (2024).

While our work is motivated by autonomous driving applications, the principle of learning per-agent relevance naturally extends to other multi-agent and multi-object settings, such as swarm robotics, multi-agent reinforcement learning, or video scene understanding, where only a subset of entities meaningfully influence a decision. By identifying and prioritizing relevant entities, similar approaches could improve scalability, reduce computational costs, and offer insights into the decision-making processes in these domains.

We introduced a reinforcement learning approach to estimate agent relevance in driving scenarios. By formulating relevance scoring as an agent-masking MDP, we enable end-to-end training of a scoring policy with a driving policy in the loop. Our method avoids costly post-hoc attribution and repeated forward passes, making it well suited for real-time autonomy stacks. In closed-loop evaluation, we show that comparable driving performance can be achieved while processing an order of magnitude fewer agents, highlighting the benefits of our approach in terms of behavior model introspection and dynamic compute allocation.

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

# A  HYPERPARAMETER SENSITIVITY

For each key hyperparameter, we ran a one-dimensional sweep using $0.1, 1, 10$ the default value while keeping all others fixed. The resulting learning curves show that performance degrades gracefully under both under- and over-scaled settings, and the method remains stable across a broad range, indicating limited sensitivity of RDAR to loss hyperparameter tuning.

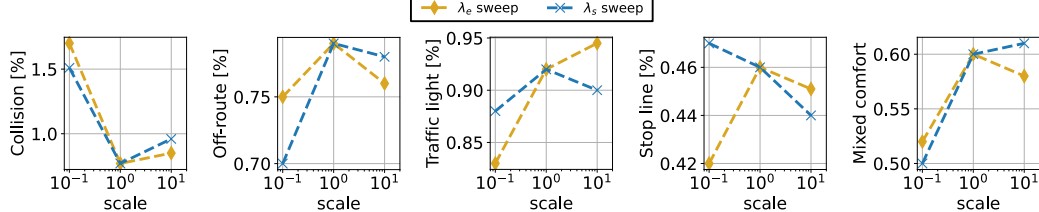

Figure 8: Effect of sweeping the loss hyperparameters by two orders of magnitude. The performance degrades gracefully. As expected, a higher smoothing weight $\lambda_s$ leads to higher comfort score, but also higher collision rate.

# B  ADDITIONAL QUALITATIVE EXAMPLES

In this appendix we provide additional qualitative examples of RDAR's performance in assessing agent relevance.

**Consistency in time**  The consistency of the relevance scores across different time steps is an important feature of RDAR, and is controlled by the smoothing loss defined in equation 10. Fig. 9 provides these visualizations. The sequence in Fig. 9a is particularly notable due to the highly cluttered scene.

**Failure examples**  One pattern we observed in the way RDAR attributes relevance is that vehicles parked on the side of the road tend to be considered highly relevant (see for instance Figs. 7b and 9d). This is also confirmed by the attribution-based baseline, and would be interesting to investigate further. In fact, arguably, human drivers don't pay much attention to parked vehicles.

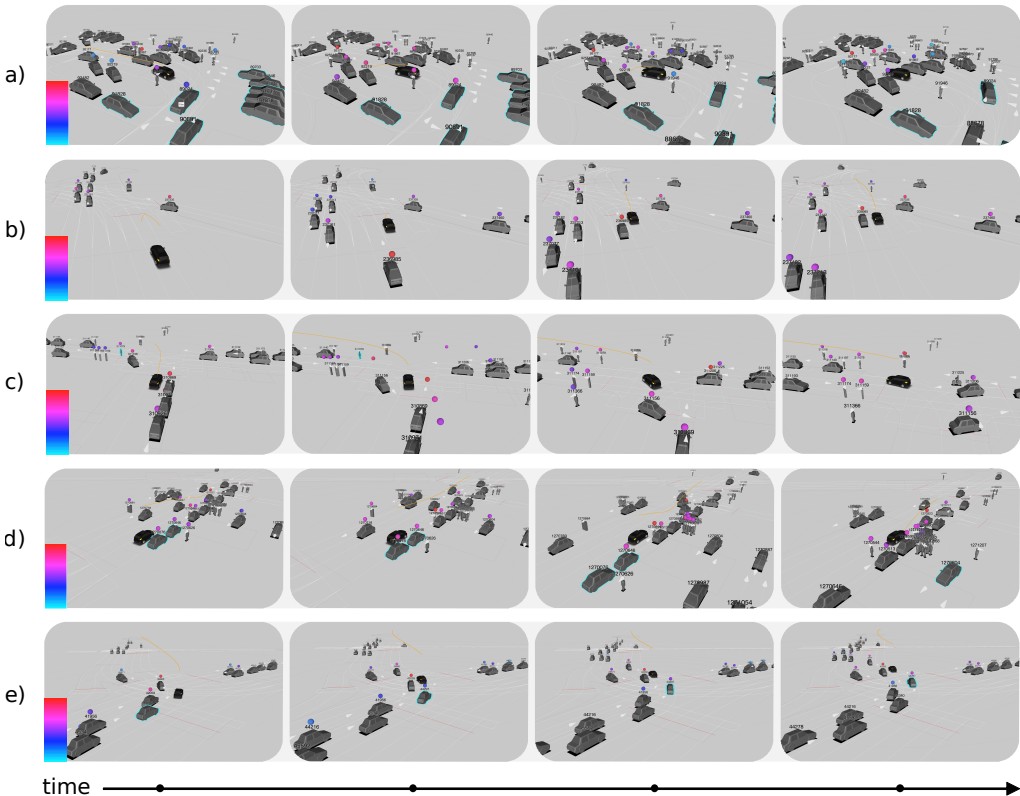

Figure 9: Snapshots in time showing the qualitative evolution of agent relevance in diverse scenarios. a) left turn in heavy traffic, b) fast car overtaking on the left while approaching an intersection, c) left turn into a two way street, d) driving straight with person getting out of a parked vehicle, e) left turn at busy intersection. The top-$k$ relevant agents are labeled with a colored dot hovering over them. As shown by the scale in the bottom left of the first snapshots of each sequence, red corresponds to higher relevance and light blue corresponds to lower relevance.

