# OpenReview forum: "RDAR: Reward-Driven Agent Relevance Estimation for Autonomous Driving"
_ICLR.cc/2026/Conference — Submitted to ICLR 2026_

### Official Review · Reviewer_cdyS · 2025-10-29

**Soundness:** 3
**Presentation:** 2
**Contribution:** 2
**Rating:** 4
**Confidence:** 4

**Summary:**

This paper proposes RDAR , an RL-based framework for estimating which surrounding agents truly influence the ego vehicle’s behavior in autonomous driving. Instead of processing all scene entities with quadratic attention cost, RDAR formulates relevance estimation as an MDP, where actions are binary masks selecting agents passed to a frozen driving policy. A scoring policy is trained in closed-loop using off-policy actor–critic with V-trace, optimizing standard driving rewards (safety, progress). At inference, the top-k most relevant agents are retained, reducing input complexity. Evaluated on large-scale real-world data across several cities, RDAR maintains nearly identical driving performance to the full model while processing far fewer agents, achieving O(1) efficiency compared to O(N) attribution methods.

**Strengths:**

The paper proposes a conceptually novel framing of agent relevance estimation as an RL-based agent masking problem. RDAR offers substantial computational savings by enabling per-agent filtering in one forward pass.

The paper is well-structured. The method is designed to complement existing driving policies, which offers generalizable possiblilities. Visualizations demonstrate that RDAR’s learned relevance maps correspond well with human intuition

**Weaknesses:**

The main innovation lies in problem framing rather than algorithmic advancement. The paper states RDAR uses the same reward as the driving policy, but does not provide its explicit form or show ablations on reward sensitivity.

Experiments are confined to one in-house dataset. It remains unclear whether RDAR generalizes to other planners (e.g., learning-based or rule-based), datasets (e.g., nuPlan, WOMD), or domain shifts.

Although results are strong on average, the paper lacks qualitative or quantitative analysis of cases where RDAR incorrectly masks critical agents

The training requires large-scale distributed RL (IMPALA-like), but details on computational cost, convergence time, and sample efficiency are not reported.

The paper does not include comparisons or discussion of recent interaction reasoning frameworks such as JFP or braid-based, which explicitly model multi-agent dependencies.

**Questions:**

How sensitive is RDAR’s performance to the definition of the driving reward?

If the frozen driving policy πᴰ is changed (e.g., a smaller or less robust model), can RDAR be reused or fine-tuned efficiently, or must it be retrained from scratch?

How does RDAR scale in dense scenes with hundreds of agents? Does the learned relevance distribution degrade or saturate under heavy traffic?

How does RDAR compare to lightweight attention sparsification techniques (e.g., Performer, linear attention) in accuracy and compute trade-offs?

---

> ### Author Response · Authors · 2025-12-03
>
> We thank the reviewer for the insightful and constructive comments. We address each comment below (comments about testing on public datasets have been addressed in the general response to reviewers at the top):
>
> - The paper states RDAR uses the same reward as the driving policy, but does not provide its explicit form or show ablations on reward sensitivity.
>
> Unfortunately, the exact form of the driving reward is proprietary knowledge and we are not able to include it in the paper. The driving reward, as in many other cases in the autonomous driving case, is a combination of progress, safety, lane keeping, collision avoidance with static objects and other agents. However, we would like to emphasize that our framework is designed to act ON TOP of specific autonomous driving frameworks and their corresponding reward functions. The driving reward is used in components (7) and (8) of the loss function. We added some comments on the driving reward at the end of the “Reinforcement learning framework” paragraph.
> We also added a hyperparameter sensitivity analysis in Appendix A.
>
> - Although results are strong on average, the paper lacks qualitative or quantitative analysis of cases where RDAR incorrectly masks critical agents.
>
> We added some discussion about some observed patterns of incorrect masking in Appendix B. In particular, vehicles parked on the side of the road seem to be considered highly relevant both by RDAR and the attribution baseline.
>
> - details on computational cost, convergence time, and sample efficiency are not reported
>
> We added more information on the computational cost and training times in the results section. We also added some example training curves in a new Fig. 5.
>
> - The paper does not include comparisons or discussion of recent interaction reasoning frameworks such as JFP or braid-based, which explicitly model multi-agent dependencies.
>
> The main difference between our approach and JFP or braid-based approaches is the fact that we don’t explicitly model agent interactions, and use a transformer-based architecture. In principle, it would be possible and interesting to use architectures such as the one proposed in JFP or braid-based prediction works and see how the specific model architecture affects performance. In the present work, we focused on the relevance scoring task and its RL formulation. Even if we provide some ablation studies on model architecture and different amounts of information used, our main contribution is not the model architecture, but the novel problem formulation and training framework, which is architecture agnostic. JFP or braid-based models could be used as well.
>
> - If the frozen driving policy πᴰ is changed (e.g., a smaller or less robust model), can RDAR be reused or fine-tuned efficiently, or must it be retrained from scratch?
>
> We thank the reviewer for the interesting question. The answer depends on the actual use of the RDAR model. If the goal is to use such relevance scoring model for introspection purposes, then the RDAR model should be trained in the loop with the actual policy one is interested in introspecting.
> If instead the goal is to use RDAR e.g. for online compute prioritization, then the model can be reasonably transferred. In fact, the learned scoring policy outputs per-agent scores given a driving scene. If the underlying driving policy is fragile, however, adding RDAR in the loop will likely not robustify it. In fact, RDAR is an agent scoring model, and does not operate on other parts of the input such as road information.

---

> > ### Author Response · Authors · 2025-12-03
> >
> > - How does RDAR scale in dense scenes with hundreds of agents? Does the learned relevance distribution degrade or saturate under heavy traffic?
> >
> > The RDAR model, as the underlying driving policy, processes up to the closest 128 agents. Any agent beyond the closest 128 will not be considered by the RDAR model or the driving policy. This is a design decision justified by achieved driving performance and diminishing returns. We did not observe any degradation in performance in the presence of ~100 agents. We added some examples of heavily cluttered scenarios in appendix B.
> >
> > - How does RDAR compare to lightweight attention sparsification techniques (e.g., Performer, linear attention) in accuracy and compute trade-offs?
> >
> > Attention sparsification techniques such as Performer or linear attention are methods aimed at reducing the number of FLOPs required by the attention mechanisms. This is typically done through approximations that scale linearly with the number of tokens instead of quadratically. In our case, the goal is to learn a measure of importance for the input tokens (particularly, agents in a driving scene). This is something that is not addressed by attention sparsification techniques, which mostly focus on low level computation and approximation, rather than reasoning about token importance.
> > One work that is related to ours in terms of goals is mixture-of-depths. The paper proposes a strategy to automatically learn skip connections in a transformer, such that attention is performed on a subset of the input tokens. The goal in this case is compute prioritization, which is also the goal of RDAR. We added this reference in the discussion section.

---

### Official Review · Reviewer_vX6B · 2025-10-31

**Soundness:** 2
**Presentation:** 2
**Contribution:** 2
**Rating:** 2
**Confidence:** 3

**Summary:**

This paper presents RDAR (Reward-Driven Agent Relevance), a reinforcement learning framework for estimating the relevance of surrounding agents in autonomous driving scenarios. Experiments on large-scale urban datasets from cities such as Las Vegas, Seattle, and San Francisco demonstrate that RDAR can achieve similar levels of safety, comfort, and progress while processing only about ten agents per frame, leading to an order-of-magnitude reduction in computational complexity. Although the method is technically well-executed and produces interpretable results, its main contribution lies in applying existing reinforcement learning tools to a specific application domain rather than introducing a new theoretical framework. As a result, while the work offers practical impact for real-time autonomous driving, it may be regarded as too applied and incremental for ICLR’s main track standards.

**Strengths:**

The paper’s primary strength lies in the clarity and practicality of its idea. RDAR provides a simple yet effective method for reducing computational overhead in autonomous driving models by explicitly learning per-agent relevance through reward feedback. The formulation as an MDP is clearly presented, and the training procedure--combining the Gumbel-top-k trick for sampling and an actor–critic framework--is described in a reproducible and technically sound manner. The experiments are extensive, using multiple quantitative metrics such as collision rate, off-road violations, comfort, and progress, all of which support the effectiveness of the proposed method. The visualizations of relevance scores are also intuitive and align well with human reasoning about which agents matter most in driving decisions, adding interpretability to the model’s behavior. Furthermore, the method has clear industrial relevance, as it can be integrated into real-time driving stacks to prioritize computational resources dynamically and improve inference efficiency. Overall, the paper succeeds in demonstrating a conceptually simple but practically impactful contribution to the field of autonomous driving.

**Weaknesses:**

Despite its practical utility, the paper’s novelty and theoretical depth are limited. The central idea--using reinforcement learning to optimize a masking policy that filters input agents--is relatively straightforward and can be seen as an applied variant of existing work in reinforcement-learning-based feature selection. The paper does not introduce new algorithmic components, theoretical insights, or analyses that would substantiate it as a methodological breakthrough. Moreover, the work is tightly coupled to the autonomous driving domain, relying on proprietary datasets and simulators, and lacks evidence of generalizability to other contexts. The proposed system assumes access to a high-quality, frozen driving policy, meaning that RDAR’s success depends heavily on the underlying planner’s design rather than the novelty of the relevance model itself. Importantly, there is no theoretical justification or analysis that explains why reward-driven relevance estimation should outperform attention or attribution-based alternatives, nor any guarantee regarding optimality or convergence. Consequently, while the results are compelling, the paper does not provide sufficient conceptual depth or generality to meet the expectations of ICLR’s main track, which prioritizes fundamental advances in machine learning theory and algorithms. The work would likely be more appropriate for conferences such as CoRL, IROS, or ITSC, or for specialized NeurIPS or ICLR workshops focusing on real-world reinforcement learning or autonomous driving applications.

**Questions:**

Several questions arise regarding the stability, generality, and theoretical grounding of the proposed approach. It would be useful to know whether the learned relevance scores remain consistent over time and across scenes, or whether they fluctuate as agents enter and leave the field of view. Another question concerns the generalizability of RDAR: can the framework be applied to other multi-agent or multi-object systems, such as swarm robotics or video understanding, without substantial retraining? Additionally, the method currently requires setting a fixed number of selected agents, k, could this threshold instead be learned adaptively as part of the policy? Since the approach is reward-driven, it would also be valuable to assess its robustness to variations in the reward function or driving policy, as both could alter the learned relevance distribution. Finally, a deeper theoretical connection between the reward-driven relevance policy and established notions such as value-function-based importance weighting or causal influence would strengthen the contribution and situate the work more firmly within reinforcement learning theory.

---

> ### Author Response · Authors · 2025-12-03
>
> We thank the reviewer for their constructive feedback. We address each individual comment below (comments about testing on public datasets have been addressed in the general response to reviewers at the top):
>
> - [This work] can be seen as an applied variant of existing work in reinforcement-learning-based feature selection
>
> We thank the reviewer for the pointer on feature selection. We found our work to be different from RL-based feature selection. In extreme synthesis, as opposed to RL-based feature selection, which applies RL to select important features for a model in a supervised learning task, we deal with coming up with a “feature selection” mechanism for a sequential decision making problem (i.e. driving). We  added a paragraph in the related works section drawing the comparison between our method and RL-based feature selection.
>
> - [This work] does not introduce algorithmic components, theoretical insights or analysis that would substantiate it as a methodological breakthrough
>
> Although we can probably say that RDAR does not qualify as a "breakthrough" we respectfully disagree with this comment. This is the first time that agent ranking in autonomous driving is formulated as an MDP, and a ranking policy is obtained without any ground truth data. The conceptual novelty, as recognized by the other reviewers, lies in applying multi-armed bandit-like methods to an intrinsically continuous, dynamic, sequential field such as autonomous driving. Therefore, we do believe the idea is novel enough to qualify for publication at a top tier venue such as ICLR.
>
> - [This work] lacks evidence of generalizability to other contexts
>
> Our implementation happens to run inside a proprietary autonomous driving stack, but the method itself is not tied to any proprietary components: it only requires standard state observations and generic reward terms (lane-keeping, collision avoidance, progress), which are available in any driving simulator. Importantly, our approach does not rely on heuristics, map-specific features, or proprietary perception modules. We emphasize that the contribution is an algorithmic framework that is simulator-agnostic and applicable to any RL-based autonomous driving setup.
>
> - The proposed system assumes access to a high-quality, frozen driving policy
>
> This is inaccurate. The “success” of RDAR should be evaluated as the ability to identify the agents that A SPECIFIC POLICY finds relevant. This is useful for introspection, interpretability and model improvement, and for compute prioritization for the specific policy. If the underlying policy is low-quality, the RDAR model will likely learn relevance scores biased by spurious correlations, and allow to better identify the spurious correlations in a low quality policy.
>
> - No theoretical justification or analysis that explains why reward-driven relevance estimation should outperform attention or attribution-based alternatives
>
> Attention-based relevance estimation has been found in previous work to not be very effective (Attention is not explanation by Jain and Wallace). Therefore, we did not consider methods based on attention weights.
> Attribution methods adopt a leave-one-out approach, and measure the effect on the predicted action. There are two principled reasons for why RDAR should be preferred to attribution. 1) if two agents lead to the same action (i.e., two agents with the right of way at an intersection), leaving either out will lead to very low action variation, and they will be incorrectly assigned low importance. 2) RDAR is more computationally efficient. It requires one pass, while attribution methods require N passes. This is critical in real-time applications.
>
> - There is not any guarantee regarding optimality or convergence
>
> This is true, we do not provide any theoretical guarantees on optimality or convergence. However, we argue that most deep learning approaches do not either, and mostly rely on empirical validations. We believe, as it is also observed by the other reviewers, that our empirical results show strong evidence of the effectiveness of RDAR.

---

> > ### Author Response · Authors · 2025-12-03
> >
> > - It would be useful to know whether the learned relevance scores remain consistent over time and across scenes, or whether they fluctuate as agents enter and leave the field of view.
> >
> > The degree of consistency in time is controlled by the regularization part of the reward function (smoothing loss in equation (10)). This requires empirical tuning. We added some sequential snapshots to show relevance consistency over time in appendix B.
> >
> > - Can the framework be applied to other multi-agent or multi-object systems, such as swarm robotics or video understanding, without substantial retraining?
> >
> > A relevance model trained on autonomous driving data is unlikely to transfer zero-shot to a different domain, nor should it be expected to. However, we do believe that the key ideas and insights, and the general methods developed for RDAR can naturally be applied to other settings in which multiple objects or agents need to be ranked and there is no access to ground truth labels.
> >
> > - The method currently requires setting a fixed number of selected agents, k, could this threshold instead be learned adaptively as part of the policy?
> >
> > Thank you for raising this interesting issue. We would like to first clarify that the method does not require fixing k in advance. In fact, at training time, k is randomized per scenario. We found that this encourages learning truthful relevance scores, instead of just differentiating between top-k and not top-k agents. At each timestep, the RDAR model assigns a score to all agents in the scene. This can effectively be interpreted as a distribution whose logits are the relevance scores. The sampling procedure used can be a variable k sampling (one example could be top-p, popular in large language models). At the same time, having a fixed k allows to maximize usage of available resources. If for example we use the RDAR model to compute vision embeddings on a subset of agents, using the largest fixed k possible would be the best strategy. Learned strategies to pick k based on the complexity of the scene would be interesting to explore, but go beyond the scope of this work. We added some discussion about this in the “training details” paragraph of the “Implementation” section.
> >
> > - Since the approach is reward-driven, it would also be valuable to assess its robustness to variations in the reward function or driving policy, as both could alter the learned relevance distribution.
> >
> > RDAR being robust to variations in the driving policy would be a bug, and not a feature. The value of the RDAR model is being able to capture the notion of relevance underlying the specific driving policy used. Assume, for example, we have a driving policy that never turns right. Then, all the pedestrians crossing on the right side of an intersection are irrelevant. In the same way, a more aggressive policy defines a different notion of relevance compared to a conservative policy.
> >
> > - A deeper theoretical connection between the reward-driven relevance policy and established notions such as value-function-based importance weighting or causal influence
> >
> > We added some comments on the meaning of the value function in our setting at the end of page 5. About causal influence, we believe that using the driving reward directly for training a relevance model, instead of for example using post-hoc methods such as attribution, helps with avoiding spurious correlations. Indeed, in our case the relevance of an agent is estimated taking into account 1) the reward and 2) the system’s dynamics (i.e. the temporal aspect).

---

### Official Review · Reviewer_RB4p · 2025-11-01

**Soundness:** 3
**Presentation:** 3
**Contribution:** 2
**Rating:** 4
**Confidence:** 3

**Summary:**

This paper proposes RDAR, a reinforcement-learning method that learns per-agent relevance in autonomous driving by masking out non-critical agents from a frozen driving policy’s input. The relevance estimator is trained as an MDP with reward signals from driving performance. Experiments on large-scale driving data show comparable safety and comfort while processing far fewer agents.

**Strengths:**

1. RDAR improves computational efficiency: a single-pass O(1) scoring step replaces O(N) attribution, and with k≈10 it matches full-input safety/progress while delivering quadratic savings when downstream attention is pairwise.

2. Relevance heatmaps match human intuition (e.g., yield/stop cases), making debugging easier and supporting targeted compute allocation.

**Weaknesses:**

1. Experiments rely on a non-public large-scale driving dataset, preventing others from reproducing or benchmarking RDAR on open platforms (e.g., Waymo, nuScenes).
2. The scoring policy’s quality is tied to the frozen driving model and reward design; biases or suboptimal rewards in the base system could propagate to RDAR’s relevance learning.
3. A constant number of selected agents may not suit varying scene complexity; adaptive or learnable selection would better reflect real-world driving conditions.
4. Although inference is efficient, the closed-loop RL training likely requires heavy simulation and distributed resources, yet the paper provides no runtime or resource metrics.

**Questions:**

1. Can k be dynamically adapted per scenario?
2. How stable is agent selection over time?
3. Any plan to test on public datasets ?
4. Can the resource savings be quantitatively measured?

---

> ### Author Response · Authors · 2025-12-03
>
> We thank the reviewer for the constructive feedback, and for recognizing the strengths of our work. We address each comment below (the comments about testing on public datasets have been addressed in the general response to reviewers at the top):
>
> - The scoring policy’s quality is tied to the frozen driving model and reward design; biases or suboptimal rewards in the base system could propagate to RDAR’s relevance learning.
>
> We do not believe the fact that the RDAR model is tied to the frozen driving model is a weakness. It is, on the other hand, one of the main features of RDAR, as the notion of relevance is intrinsically tied to a driving policy. This is also true for human drivers. As an example, aggressive drivers need to pay attention to different agents compared to more unassertive drivers. Similarly, assume we have a driving policy that never turns right. Then, all the pedestrians crossing on crosswalks on the right side of the road are irrelevant.
> The goal is to estimate the relevance WITH RESPECT to a specific driving policy, which allows to introspect the model, identify bad behaviors and correlations, and prioritize compute resources.
>
> - A constant number of selected agents may not suit varying scene complexity; adaptive or learnable selection would better reflect real-world driving conditions.
>
> We added some comments on different agent sampling strategies in Section 4. In short, k does not have to be fixed, and different sampling strategies can be used. One option is top-p, widely used for next token sampling in LLMs. Other strategies varying k depending for instance on scene complexity, could be explored in future work.
>
> - Although inference is efficient, the closed-loop RL training likely requires heavy simulation and distributed resources, yet the paper provides no runtime or resource metrics.
>
> We added details on the resources used, as well as training curves showing the sample efficiency of our approach.
>
> - Can k be dynamically adapted per scenario?
>
> We added some comments on different agent sampling strategies in Section 4. In short, k does not have to be fixed, and different sampling strategies can be used. One option is top-p, widely used for next token sampling in LLMs. Other strategies varying k depending for instance on scene complexity, could be explored in future work.
>
> - How stable is agent selection over time?
>
> The agent selection over time is stable, and is mainly controlled by the smoothing component of our loss function (i.e., equation 10). We added some snapshots in time to show the consistency of the scores in appendix B.
>
> - Can the resource savings be quantitatively measured?
>
> The resource savings depend on the actual hardware used. We did not run this model on our real-world vehicle, which would give the most meaningful comparison given the real-time nature of our application. We did however profile inference time on the evaluation set, when running the RDAR model on A100 GPUs. A vectorized implementation of the attribution method (N passes) is 10x slower than the RDAR model (single pass). We also added a comment on this in the paper.

---

### Author Response · Authors · 2025-12-03

We proposed a Reward-Driven Agent Relevance (RDAR) estimation method that is able to learn to rank agents in a driving scene through reinforcement learning, thus not needing any labelled data. We show the effectiveness of these rankings by analyzing the behavior of a pre-trained driving policy when only processing a fraction of the surrounding agents.
We thank the reviewers for the time and effort they put into reading our paper, and for the feedback they gave us.
We are encouraged that all the reviewers acknowledge the benefits of RDAR in terms of improvements in real-time computational efficiency and debugging/introspection opportunities. We are also glad that the clarity, soundness and practicality of our ideas were recognized, and the structure of our paper was appreciated. We are pleased that Reviewer 3 (cdyS) recognizes the conceptual novelty of our ideas, which, as also Reviewer 2 (vX6B) observed, are of high practical impact.

One of the main weaknesses highlighted is the fact that we did not benchmark our approach on public datasets. We agree that this would strengthen our results and add value for the research community. The main reason why we did not do that is that our code is tightly integrated with a proprietary codebase, with its own data formats and conventions. Testing our method on other datasets would require a complete reimplementation, which is infeasible within the scope of this work. However, we would like to provide some arguments for why our results are still relevant.
- Our method does not rely on any dataset-specific assumptions, and all components are designed to be dataset-agnostic.
- The proprietary dataset used, as highlighted in the paper, is a large collection of real-world driving data from multiple cities in the United States that is used to train and evaluate production-level driving models. The dataset includes a broader distribution of scenarios compared to public datasets (roughly 10x total driving hours compared to Waymo Open Dataset or Argoverse).
- We provide details and analysis that fully define our method, such that anyone interested in using our method paired with a different planning algorithm and corresponding cost/reward functions is able to do that.

As the reviewers observed, we apply a simple yet powerful idea to autonomous driving, and discuss its performance. The value and goals of this paper lie in the conceptual novelty applied to an underexplored aspect of autonomous driving, rather than an improvement over state-of-the-art methods, which we agree would require more extensive benchmarking on public resources.

We respond to each individual comment below. We also uploaded a revised version of the manuscript with changes highlighted in blue.

---

### Meta-Review · Area_Chair_9mnz · 2026-01-04

**Summary:**

Across the reviews, the paper is consistently described as clear, well-structured, and practically motivated. All reviewers agree that RDAR formulates agent relevance estimation in autonomous driving as a reinforcement learning problem and that it can substantially reduce computational cost by selecting a small subset of surrounding agents for a frozen driving policy. Reviewers also acknowledge that the learned relevance scores are often intuitive and useful for debugging and introspection, and that the approach has clear industrial applicability. However, the primary concerns raised by reviewers are substantial and remain largely unresolved. Most critically, all experimental results rely exclusively on a proprietary dataset and simulator, preventing reproducibility, independent validation, and fair comparison against existing methods on standard benchmarks. In addition, multiple reviewers characterize the contribution as primarily an application of existing reinforcement learning ideas to a specific autonomous driving setting, with limited algorithmic novelty, theoretical depth, or generality. The method is also tightly coupled to a frozen driving policy and its reward design, further limiting the scope of the contribution. While the rebuttal provides clarifications, additional discussion, and some implementation details, it does not materially address these core limitations.

**Reviewer Concerns:**

Concerns addressed by the rebuttal:
1. Computational efficiency and profiling (RB4p, cdyS): The authors add inference-time profiling and training resource details, confirming that RDAR is faster than attribution-based baselines and feasible to train in their internal setup. These additions improve transparency but do not change the nature of the contribution.
2. Temporal stability of relevance scores (RB4p, vX6B): The rebuttal explains that stability is controlled via a smoothing term and adds qualitative examples in the appendix. This addresses a presentation gap but does not constitute a deeper analysis.
3. Fixed vs. adaptive k (RB4p, vX6B): The authors clarify that k need not be fixed and discuss alternative sampling strategies (e.g., top-p).
4. Dependence on a frozen driving policy: The authors explicitly state that relevance is defined with respect to a specific policy and argue this is a feature rather than a limitation. This clarification is consistent with the paper’s intent but reinforces that RDAR’s relevance notion is policy-specific rather than general.
5. Comparison to attribution and attention methods: The rebuttal argues qualitatively that attribution methods are inefficient and attention weights are unreliable.

Concerns partially addressed or still outstanding:
1. Use of proprietary data only (RB4p, vX6B, cdyS): All reviewers identify the lack of public benchmarks as a major weakness. The authors explain that reimplementation on public datasets is infeasible and argue that their internal dataset is large and realistic. Nevertheless, the absence of a public, reproducible evaluation remains unresolved.
2. Limited novelty and theoretical contribution (vX6B, cdyS): Several reviewers view RDAR as an application of existing RL-based selection or masking ideas rather than a fundamentally new method. The rebuttal emphasizes problem framing and practical impact but does not introduce new algorithmic insights or theoretical analysis.

**Reviewer Scores:**

Reviewer RB4p: Likely unchanged. While the rebuttal addresses efficiency, stability, and reporting questions, the reviewer’s concerns about proprietary data and limited contribution remain.

Reviewer vX6B: Likely unchanged. This reviewer’s rejection is grounded in a lack of theoretical depth, novelty, and generality, which the rebuttal does not resolve.

Reviewer cdyS: Likely unchanged. Although the reviewer acknowledges conceptual novelty in framing, their concerns about generalization, missing analyses, and proprietary evaluation persist.

---

### Decision · Program_Chairs · 2026-01-26

Reject